

# Comparison of the antibacterial properties of peptides from the hepatopancreas of red king crab and snow crab and development of an approach for red king crab peptide isolation from the hepatopancreas

Vladislav Molchanov[1], Alfiya Yunusova[2], Olesya Kazantseva[3], Alexander Yegorov[1], Alexander Lukin[4], Alexander Timchenko[5], Vitaly Novikov[6], Nikolay Novojilov[7] and Maria Timchenko[1]

[1] Laboratory of NMR Biosystems, Institute of Theoretical and Experimental Biophysics, Russian Academy of Sciences, Pushchino, Moscow Region, Russia

[2] Laboratory for Cell Engineering, Institute of Theoretical and Experimental Biophysics, Russian Academy of Sciences, Pushchino, Moscow Region, Russia

[3] Laboratory of Bacteriophage Biology, GK Skryabin Institute of Biochemistry and Physiology of Microorganisms, Pushchino Scientific Center for Biological Research of the Russian Academy of Sciences, Federal Research Center Pushchino, Pushchino, Moscow Region, Russia

[4] Group of Applied Enzymology, Institute for Biological Instrumentation, Federal Research Center "Pushchino Scientific Center for Biological Research of the Russian Academy of Sciences", Pushchino, Moscow Region, Russia

[5] Group of Experimental Research and Engineering Oligomeric Structures, Institute of Protein Research, Russian Academy of Sciences, Pushchino, Moscow Region, Russia

[6] Polar Branch of Russian Federal Research Institute of Fisheries and Oceanography, Murmansk, Russia

[7] MM Shemyakin and Yu A. Ovchinnikov Institute of Bioorganic Chemistry, Russian Academy of Sciences, Moscow, Russia

Corresponding author
Maria Timchenko,
maria_timchenko@mail.ru

## ABSTRACT

The hepatopancreas of crustaceans is an important immune organ involved in the secretion of immune effectors such as antimicrobial peptides. Crustacean antimicrobial peptides are very diverse and have a broad spectrum of activity, but are poorly explored. In this work, the activity of antibacterial peptides from the hepatopancreas of red king crab and snow crab was compared. Both peptides were found to be highly effective in inhibiting the growth of gram-positive bacteria. At the same time, the peptide from the red king crab was found to be highly sensitive to the presence of salt and a disulfide bond reducing agent, in contrast to the peptide from the snow crab. An approach for the chromatographic purification of the red king crab peptide from the hepatopancreas was developed. Both peptides are of interest for the development of novel antibacterial drugs.

## INTRODUCTION

The golden age of antibiotics began in the middle of the last century with the discovery and first trials of purified penicillin in the 1940s (*Lobanovska & Pilla, 2017*). However, in spite of the great hopes raised in the field of antibacterial therapy and the emergence of antibiotics that are now in their fourth generation, this progress has had unforeseen consequences. The system of bacterial defence against antibiotics has improved along with antibiotics and technology. Antimicrobial resistance (AMR) is a major threat to public health worldwide (*Ribeiro et al., 2022*).

The treatment of infections caused by microorganisms that are resistant to antimicrobials requires more expensive second- or third-line antibiotics, isolation and a longer stay in hospital. These types of infections are associated with high mortality rates. In 2019, approximately 4.95 million deaths were associated with bacterial AMR (*Antimicrobial Resistance Collaborators, 2022*). A total of 3.57 million deaths were caused by the six leading AMR pathogens—*Escherichia coli*, *Staphylococcus aureus*, *Klebsiella pneumoniae*, *Streptococcus pneumoniae*, *Acinetobacter baumannii* and *Pseudomonas aeruginosa*—identified as priority pathogens by the World Health Organization (WHO). AMR can occur as part of the process of microbial evolution (intrinsic mechanism). It can also result from the indiscriminate use of antibiotics, which leads to genetic mutations and the appearance of resistance genes that are transferable from one species to another (acquired resistance).

One of the ways to solve the problem of antibiotic resistance of bacteria is the use of antimicrobial peptides (AMPs), which are small proteins of up to 50 amino acids that differ both in nature (cationic, anionic, hydrophobic, amphipathic, enriched with disulfide bonds, cyclic, depsipeptides) and in secondary structure (from alpha helices, beta sheets, mixed alpha-beta structures to extended helices and loop structures) (*Sarkar, Chetia & Chatterjee, 2021*). The currently known AMPs (*Saucedo-Vázquez et al., 2022*) have a broad spectrum of antimicrobial activity against different microorganisms, and the diversity of their modes of action on bacteria prevents the emergence of bacterial resistance. These peptides kill bacterial cells either by disrupting the integrity of the membrane, inhibiting the synthesis of macromolecules (proteins, DNA and RNA) or by affecting metabolic enzymes. AMPs are defense molecules found in all life forms (from bacteria to mammals) and are used by unicellular organisms during interspecies interactions and by multicellular organisms as part of their innate immune system.

AMPs have a number of advantages, which makes evasion difficult for most microorganisms. Modifying amino acid residues in AMPs can alter their antimicrobial spectrum while preserving their basic biophysical properties. The development of resistance to AMP is complicated by the requirement of extensive alterations in bacterial cell membranes. Furthermore, the lack of clear molecular signatures in AMPs prevents the emergence of proteases that are specific to them (*Zasloff, 2019*). Furthermore, it was found that several tested AMPs have no cytotoxic effects on human cells (*Cuthbertson BJ & Gross, 2006*; *Ravichandran et al., 2016*; *Sivakamavalli et al., 2020*).

Due to their enormous variability, broad antimicrobial spectrum, high potency, low toxicity to eukaryotic cells and lower rate of bacterial resistance compared to conventional antibiotics, peptides are considered pharmacologically ideal in terms of safety, tolerability and efficacy profiles for the development of new therapeutics (*Zasloff, 2019*). Approximately 100 peptide-based drugs have already been approved (*D'Aloisio et al., 2021*; *Mullard, 2021*).

Marine organisms are considered a promising source of many pharmaceutical compounds, including AMPs, due to their ability to adapt rapidly to changing, often extreme, environmental conditions and to constantly fight off the effects of external viruses and pathogens. AMPs play a key role in the immune system of these organisms (*Destoumieux-Garzón et al., 2016*; *Huang & Ren, 2020*; *Smith & Dyrynda, 2015*; *Tassanakajon et al., 2018*; *Tassanakajon et al., 2010*). Currently, several families of AMPs have been identified in crustaceans (*Matos & Rosa, 2022*). In decapod crustaceans, the best studied AMP families are crustins, penaeidins and antilipopolysaccharide factors (ALFs) (*Destoumieux-Garzón et al., 2016*). Crustins, cationic proteins (6–22 kDa), are the most representative and widespread in the class *Malacostraca*, the largest class of crustaceans.

The main commercial species of crabs in the Russian Federation are decapods: the snowcrab (infraorder Brachyura) and the red king crab (infraorder Anomura). The AMP of these crabs has been little studied. However, in one study (*Haug et al., 2002*) antibacterial activity against the gram-negative microorganisms *V. anguillarum*, *E. coli* and gram-positive bacteria *C. glutamicum*, as well as weak activity against the gram-positive microorganism *S. aureus*, was found in extracts of haemolymph, haemocytes and some tissues of the red king crab. Thermal stability and proteinase resistance studies showed that red king crab antibacterial factors are proteinic in nature and also have lysozyme-like and haemolytic activity.

Three cationic cysteine (Cys)-rich peptides, named paralithocins 1–3, were also detected in haemocytes (*Moe et al., 2018*), which show moderate antimicrobial activity against several strains of marine microorganisms, but do not affect *E. coli*, *P. aeruginosa* or *S. aureus*. Screening of a cDNA library revealed a sequence of crustin, a cationic Cys-rich protein in red king crab haemocytes, encoding a mature peptide of 98 amino acids. Crustin from the red king crab was assigned to the group of type II crustins (*Sperstad et al., 2009*).

We have found a peptide (about five kDa) in the hepatopancreas extract of the red king crab (*Molchanov et al., 2023*), which hydrolyses the cell wall of *M. luteus* by cleaving the polysaccharide that is part of the peptidoglycane, as confirmed by zymography with cell wall polysaccharide and $^1$H NMR spectroscopy. The hepatopancreas extract containing the peptide effectively inhibited the growth of the gram-positive bacterium *B. tropicus* and significantly slowed the growth of *E. coli* at the initial stage of growth, and the efficacy against gram-positive bacteria was slightly higher than that of lysozyme.

AMPs of the snow crab have also been poorly studied. In *Beaulieu et al., (2009)*; *Beaulieu et al., (2010)* showed that protease treatment of snow crab waste homogenates, such as cephalothorax, hepatopancreas, led to the appearance of peptide fractions with high antibacterial activity. In *El Menif et al. (2019)* the antibacterial activity of fractions obtained from the snow crab hepatopancreas after hydrolysis with the endoprotease Protamex and concentration by solid phase extraction was tested. The obtained fractions were found to have antibacterial activity against gram-positive *L. innocua*. Using mass spectrometry,
eleven peptides (1.0–2.2 kDa) were identified that had 80% amino acid homology with four antimicrobial peptides: the crustin-like peptide from black tiger shrimp, the antimicrobial peptide GAPDH from yellowfin tuna, the antibacterial peptide Odorranain-C7 from the frog *Odorrana grahami* and a putative antibacterial peptide from the Asian ladybird *Harmonia axyridis*. Additionally, using zymography and $^1$H NMR spectroscopy, a small peptide (about three kDa) was identified in snow crab hepatopancreas extract that can hydrolyse the cell wall and cell wall polysaccharide of *M. luteus* (*Molchanov et al., 2024*).

In this work, the activity of peptides from hepatopancreas of red king crab and snow crab was studied at different pH, ionic conditions and in the presence of a reducing agent. The antimicrobial activity of the peptides was tested against gram-positive (*B. tropicus* and *B. subtilis*) and gram-negative (*E. coli*) microorganisms. In addition, a procedure was developed for the isolation of the peptide of the red king crab from the extract of the hepatopancreas.

## MATERIALS & METHODS

### Preparation of hepatopancreas (HPC) extracts

Hepatopancreas of the red king crab (*Paralithodes camtschaticus*) and snow crab (*Chionoecetes opilio*) were collected during scientific voyages in the Barents and Kara Seas on vessels of the Polar Branch of VNIRO (Knipovich PINRO) in 2023–2024. The crabs were caught in the Russian Federation's territorial waters. This was according to the scientific quota allocated to the institute and the voyage assignments. The hepatopancreas was separated and frozen on the board and delivered to the institute. Afterwards it was kindly provided for our purposes. The laboratory produced samples of crude extracts (acetone powders), which were used for research.

### Peptide extraction

Acetonitrile (HPLC grade) and trifluoroacetic acid (extra pure grade) were used for peptide extraction. Low molecular weight protein fractions, containing both proteins and peptides, were extracted from acetone powder derived from the hepatopancreas of both red king crab (*Paralithodes camtschaticus*) and snow crab (*Chionoecetes opilio*). The extraction protocols followed established methods: red king crab using method (*Molchanov et al., 2023*) and snow crab using method (*Molchanov et al., 2024*). After extraction, all samples were lyophilized and stored frozen at −20 °C until further analysis.

### Preparations of the cell wall and cell wall polysaccharide of the gram-positive bacterium *M. luteus*

The cell wall and polysaccharide of *M. luteus* were provided by the Experienced Biotechnological Plant (EBP) of IBCh RAS. The samples of cell wall and polysaccharide were obtained as described in *Molchanov et al. (2023)*. The polysaccharide was quantified by anthrone method (by glucose) (*Fales, Russell & Fain, 1961*). The sample contained polysaccharide at a concentration of 3.3 mg/mL in the following buffer: 20 mM sodium acetate, 0.5 M NaCl, pH 5.5. The buffer reagents were supplied by Sigma-Aldrich (St. Louis, MO, USA).

## Polyacrylamide gel zymography with cell wall of *M. luteus*

Zymography to detect antibacterial activity toward cell wall was performed according to the protocol using cell wall from the gram-positive bacterium *M. luteus* as substrate (*Fukushima & Sekiguchi, 2016*). Chemicals used: acrylamide, N,N'-methylenebisacrylamide, ammonium persulfate, glycine, glycerol (Amresco); SDS (Fluka); TEMED, bromophenol blue, Coomassie R-250 (Serva); β-mercaptoethanol, Triton X-100, methylene blue, Tris (Sigma-Aldrich). Hen egg-white lysozyme (HEWL) (Amresco) was used as a positive control for hydrolase activity.

Antibacterial activity was detected after renaturation (3 h, 37 °C) as described in *Fukushima & Sekiguchi, (2016)*. The presence of antibacterial activity was assessed by the absence of background staining. Peptide activity was assessed by zymography at each isolation. For the further experiments, the peptides were eluted from the bands of the zymogram. The spots with the activity in the stained 16% zymogram gel were excised, and peptides extracted as described (*Fukutomi et al., 2005*). The samples were lyophilized.

## Turbidimetric analysis of the activity of peptide samples toward *M. luteus* cell wall at different pH, ionic strength and in the presence of reducing agent

*M. luteus* cell wall hydrolysis was assessed *via* turbidimetry following the protocol in *Fukushima & Sekiguchi (2016)*. The peptide sample (1 o.e./mL) or HEWL (0.02 mg/mL) were added to three mL of cell wall suspension as controls and the samples were incubated at 37 °C. The optical density at 540 nm was measured at certain time intervals during the incubation. Activity at different pHs was tested using 25 mM Tris–HCl (pH 7.2, 9.0) and sodium phosphate buffer (pH 5.5). To prepare the phosphate buffer pH 5.5, one M solutions of $Na_2HPO_4$ and $NaH_2PO_4$ (Sigma-Aldrich) were used. To study the effect of ionic conditions, different concentrations of sodium chloride were used: 10, 50 and 150 mM. The effect of the disulfide bond reducing agent on the activity of the peptide was studied in the presence of dithiothreitol (DTT) (Sigma-Aldrich) at concentrations of one mM and 10 mM.

## Bacterial strains and growth conditions

Bacterial strains were sourced from the All-Russian Collection of Microorganisms (VKM) and the American Type Culture Collection (ATCC). Cultivation was carried out in Lysogeny Broth (LB) medium and on LB agar plates. The agar plates contained 1.5% (w/v) agar for the bottom layer and 0.5% (w/v) agar for the top layer. All cultures were incubated at 37 °C.

## Lytic activity of peptides

The lytic activity of peptides from red king crab and snow crab was evaluated against the following bacterial strains: Gram-positive bacteria *Bacillus tropicus* ATCC 4342 and *Bacillus subtilis* WB800N, as well as the Gram-negative bacterium *Escherichia coli* MG1655. Peptide solutions (0.1 o.e./ml) were tested on bacterial lawns. For this, 100 µL of the bacterial culture was mixed with four mL of soft agar and poured onto Petri dishes pre-prepared with a solid 1.5% LB agar base. The plates were left at room temperature for 30 min to solidify. Subsequently, 10 µL of each peptide solution was added to the lawn. HEWL
solution (three mg/mL) was used as a positive control, and SM+ buffer (50 mM Tris–HCl (pH 7.5), 100 mM NaCl, 1 mM $MgSO_4$, 0.01% gelatin) was used as a negative control. Bacterial cultures were incubated at 37 °C for 18 h in five mL of LB. These cultures were then diluted 1:100 in fresh LB and grown at 37 °C until an optical density ($OD_{595}$) of 0.2 was reached. For the experimental assay, 50 µL of the peptide solution was combined with 50 µL of the bacterial culture suspension. Positive controls consisted of 50 µL of HEWL solution (3 mg/mL) mixed with 50 µL of the bacterial suspension. Negative controls were prepared by mixing 50 µL of the bacterial culture with 50 µL of SM+ buffer. The 96-well plate was incubated at 37 °C in a FilterMax F5 microplate spectrophotometer (Molecular Devices, San Jose, CA, USA), with $OD_{595}$ measurements recorded every 10 min for 9 h. Growth curves were generated using GraphPad Prism version 8.4.343. Error bars on the plots represent the standard deviation, calculated from three independent trials.

### Chromatographic purification of red king crab peptide

All procedures were carried out at 4 °C. The following chromatographic media were used in the work: Blue Sepharose 6 Fast Flow (GE HealthCare, Chicago, IL, USA) and MonoQ (GE HealthCare). Both columns were equilibrated with buffer A (10 mM Tris–HCl, pH 7.5). The lyophilized extract of low molecular weight protein fractions from the acetone powder of red king crab HPC was dissolved in 400 µL milliQ water and centrifuged at 12,000 rpm for 5 min. The supernatant was loaded onto the Blue Sepharose 6 Fast Flow column (volume 5 ml). The flow rate was 0.1 ml/min. After complete loading of the sample, the column was washed with buffer A. Elution was performed with buffer A with different NaCl concentrations: 5, 10, 20, 30, 40, 50 mM. After purification on Blue Sepharose 6 Fast Flow, all fractions were lyophilized. The lyophilizates were dissolved in 50 µl of distilled water. Seven µl were taken for analysis of fraction composition by 16% SDS-PAGE (*Fales, Russell & Fain , 1961*) and seven µL were taken for analysis of the occurrence of antibacterial activity in the fractions by zymography with cell wall (*Fukushima & Sekiguchi, 2016*). The fraction of unbound proteins was reloaded onto the column Blue Sepharose 6 Fast Flow and eluted with increasing NaCl concentration. The fractions were lyophilized and analyzed as described above. In addition, the fraction eluted in 10 mM NaCl after the first purification and exhibiting activity in the zymogram was loaded onto the MonoQ column. The fractions were eluted with a NaCl gradient and analyzed by 16% SDS-PAGE. The activity of the fractions was evaluated by turbidimetric analysis of the cleavage of the cell wall under the action of the peptide, if it was present in the fractions.

The experiments were independently repeated >3 times (including extraction and analysis). The results of three separate experiments were averaged for the turbidimetric analysis data presented. Error bars represent standard deviation.

## RESULTS

### Comparison of the activity of peptides from the red king crab and the snow crab at different pH values

We have previously investigated the activity of the extract of low molecular weight protein fractions from the HPC of the red king crab at different pH values (*Molchanov et al., 2023*).

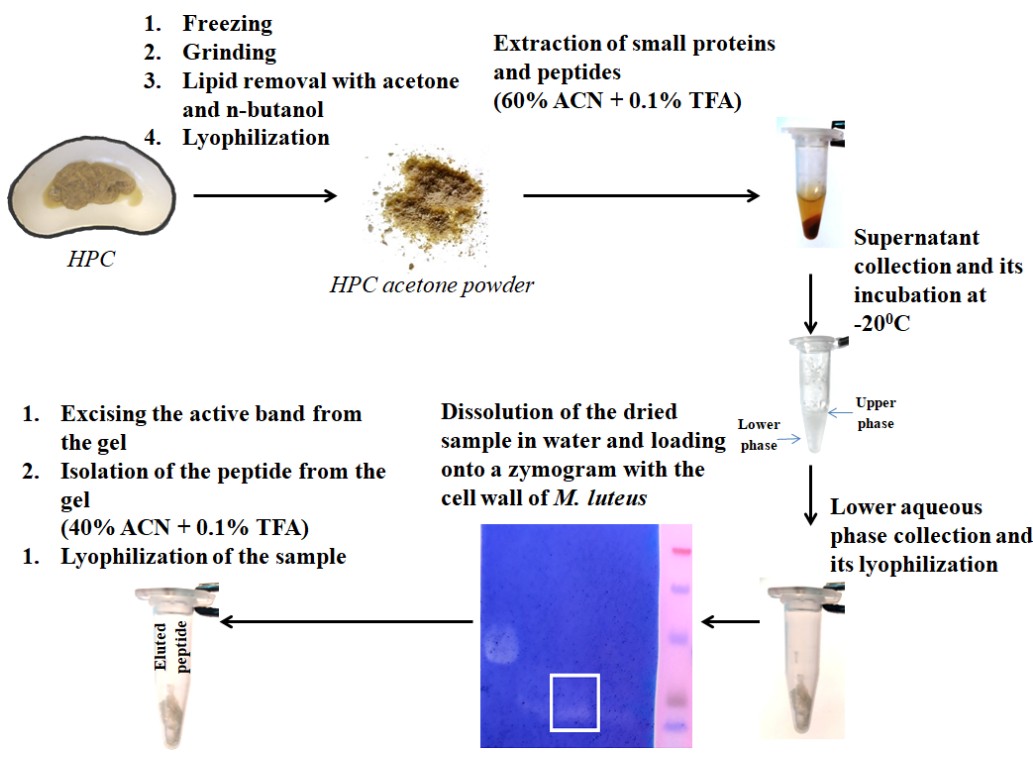

**Figure 1  Peptide extraction scheme.**

In the present paper we compared the activity of peptide preparations isolated from the gel bands with activity of red king crab and snow crabs at different pH. HEWL was used as a control for cell wall degradation. The general scheme for isolating the peptides used in the studies is shown in Fig. 1. The obtained results are presented on Fig. 2. The analysis of kinetics of cell wall degradation reveals that both peptides exhibit maximum activity at pH 7.2, which is close to HEWL. Incubation for one hour at pH 9.0 showed that red king crab peptide is less active than snow crab peptide and HEWL. After 2 h of incubation, the activity of all three samples became nearly the same. Both peptides have the lowest activity at pH 5.5.

## Investigation of the influence of ionic strength and disulphide reducing agents on the activities of peptides from red king crab and snow crab

AMPs are known to be very sensitive to salt content (*Yu et al., 2011*; *Bals et al., 1998*). The effect of different concentrations of NaCl on the activity of the peptides was analyzed. The obtained data are shown in Fig. 3. It was found that NaCl at any concentration reduced the activity of the red king crab peptide, whereas the activity of the snow crab peptide even slightly increased in the presence of salt, as with HEWL.

It is known that many AMPs have cysteine-rich regions (*Moe et al., 2018*; *Sperstad et al., 2009*) and are therefore very sensitive to the action of a reducing agent. The effect of a reducing agent of disulfide bonds, dithiothreitol, on activity was studied. The activity was

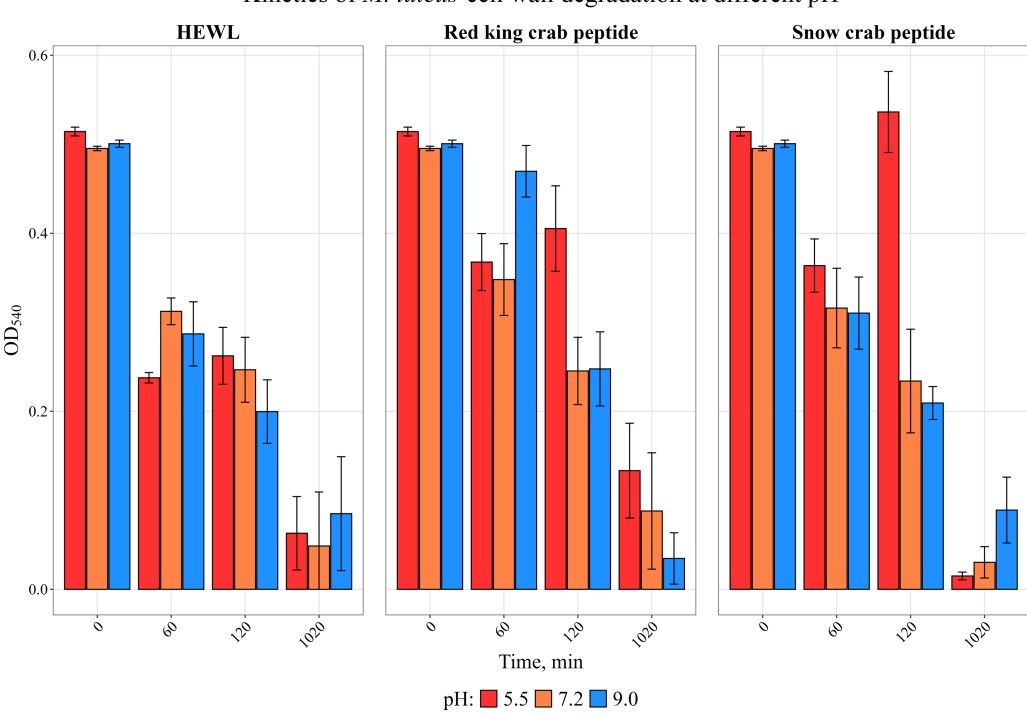

**Figure 2** **Kinetics of *M. luteus.* cell wall degradation at different pH levels in the presence of HPC peptides from red king crab, snow crab and HEWL.** Error bars show the standard deviation of OD_540.

assessed by measuring the change in optical density of the cell wall suspension at 540 nm during incubation with the peptide at 37 °C at pH 7.2 at DTT concentrations of 1 and 10 mM. It was shown (Fig. 4) that the presence of DTT strongly suppressed the activity of the red king crab peptide, whereas it had almost no effect on the snow crab peptide, as with HEWL, and even slightly increased the activity of the snow crab peptide during the first hours of incubation.

## Lytic activity of peptides

The lytic activity of peptides from red king crab and snow crab was assessed on bacterial lawns, where both peptides demonstrated pronounced lytic activity against Gram-positive *Bacillus* species (Figs. 5A–5C). Notable zones of lysis appeared on the lawns of *Bacillus tropicus* ATCC 4342 and *Bacillus subtilis* WB800N at the areas where the peptides were applied (Figs. 5B and 5C). These findings confirm the peptides strong bactericidal effects against these Gram-positive bacteria. In comparison, HEWL exhibited activity exclusively against *Bacillus subtilis* WB800N (Fig. 5C), highlighting the strain-specific susceptibility of *Bacillus subtilis* to HEWL. No lytic activity from either peptides or HEWL was observed on the lawn of the Gram-negative strain *Escherichia coli* MG1655 (Fig. 5A).

Further investigation revealed that the peptides from red king crab and snow crab demonstrate potent lytic effects against the Gram-positive *Bacillus tropicus* ATCC 4342 and *Bacillus subtilis* WB800N strains (Figs. 5E and 5F), resulting in complete inhibition

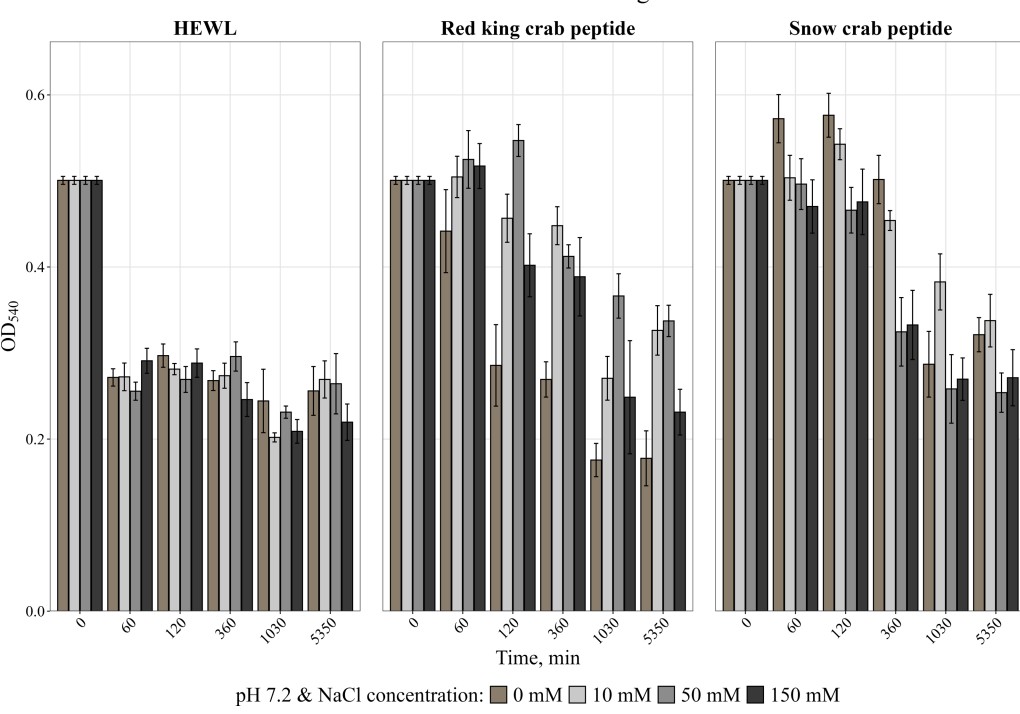

Figure 3 **Effect of different NaCl concentrations on the degradation of the cell wall of *M. luteus* by the red king crab peptide, HEWL and the snow crab peptide.** Error bars show the standard deviation of $OD_{540}$.

of bacterial growth in liquid medium. These results underscore the strong antimicrobial potential of these peptides against certain Gram-positive bacteria, consistent with the findings in the agar-based lawn assay. However, as observed in the previous experiment with agar lawns, the peptides showed no lytic activity against the Gram-negative *Escherichia coli* MG1655 (Fig. 5D). Interestingly, rather than inhibiting growth, the peptides seemed to promote the growth of *E. coli* in the liquid medium. This phenomenon could be explained by the bacteria utilizing the peptides as a source of nutrients or energy, thereby enhancing their metabolic activity.

In contrast to the peptides, HEWL, which is often used as a reference antimicrobial agent for its ability to degrade peptidoglycan in bacterial cell walls, exhibited only a modest inhibitory effect on the growth of *Escherichia coli* MG1655, as compared to the positive control (Fig. 5D). The minimal impact of HEWL on *E. coli* may be due to the species' intrinsic resistance mechanisms or the inability of HEWL to effectively penetrate the outer membrane of Gram-negative bacteria. Regarding *Bacillus tropicus* ATCC 4342, HEWL only temporarily inhibited bacterial growth, suggesting the emergence of resistant mutants during the culture process (Fig. 5E). This transient effect may indicate a partial resistance of the strain to HEWL, which could be due to natural variation in the strain or specific adaptive mechanisms developed under the experimental conditions. Interestingly, for *Bacillus subtilis* WB800N, HEWL exhibited a lytic effect comparable to that of the tested

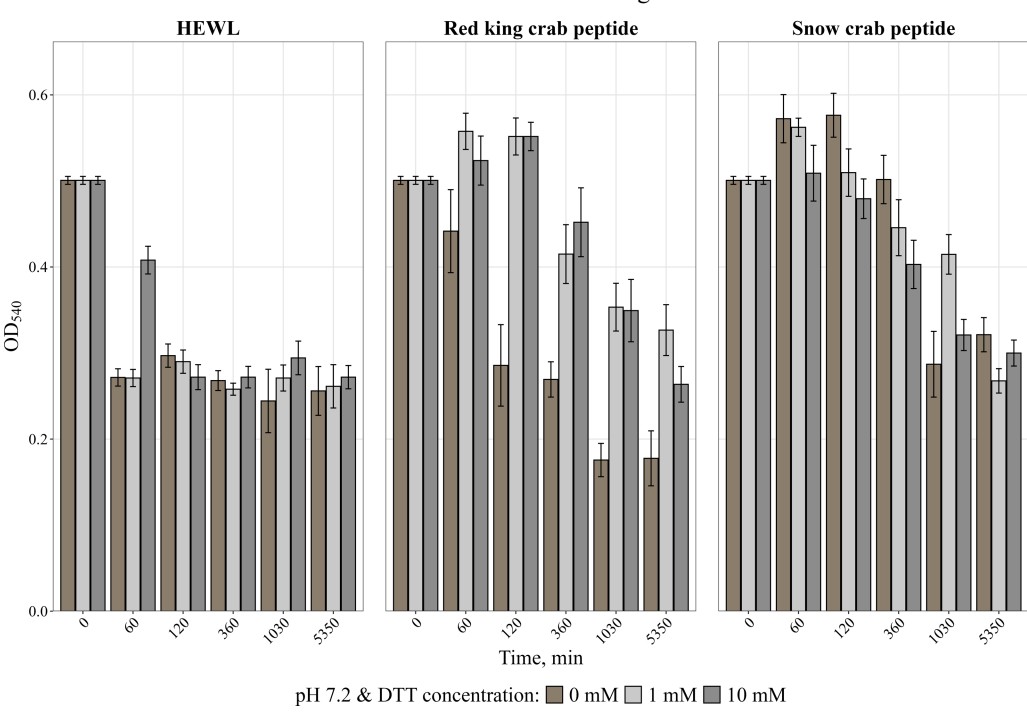

Effect of different DTT concentrations on the digestion of *M. luteus* cell wall

**Figure 4  Effect of different DTT concentrations on the degradation of the cell wall of *M. luteus* by the red king crab peptide, HEWL and the snow crab peptide.** Error bars show the standard deviation of $OD_{540}$.

peptides, making it clear that this strain is highly susceptible to both HEWL and the peptides (Fig. 5F). The results from both the agar-based and liquid culture experiments highlight the significant antimicrobial potential of the red king crab and snow crab peptides against specific Gram-positive bacteria, while also providing insight into the resistance mechanisms and susceptibility profiles of different bacterial strains.

## Development of an approach for the chromatographic purification of an antibacterial peptide from an extract of low molecular weight protein fractions from the HPC of the red king crab

It is important to obtain a pure preparation of the antibacterial peptide in order to analyze its cytotoxic properties and to identify the amino acid sequence of the peptide. Our previous attempts to purify the peptide from the extract of low molecular weight protein fractions of the red king crab hepatopancreas using gel filtration on Sephadex G-75 and Sephacryl S-200 HR did not result in any significant purification of the peptide from other proteins, although a peptide band in the five kDa region appeared on SDS-PAGE. The assumption of affinity binding to heparin-sepharose due to affinity for the polysaccharide was also not confirmed. Zymographic analysis revealed good binding of the peptide to the heterocyclic dye methylene blue, which may be due to hydrophobic interactions. It has three basic rings: two aromatic carbon rings and one containing nitrogen and sulphur.

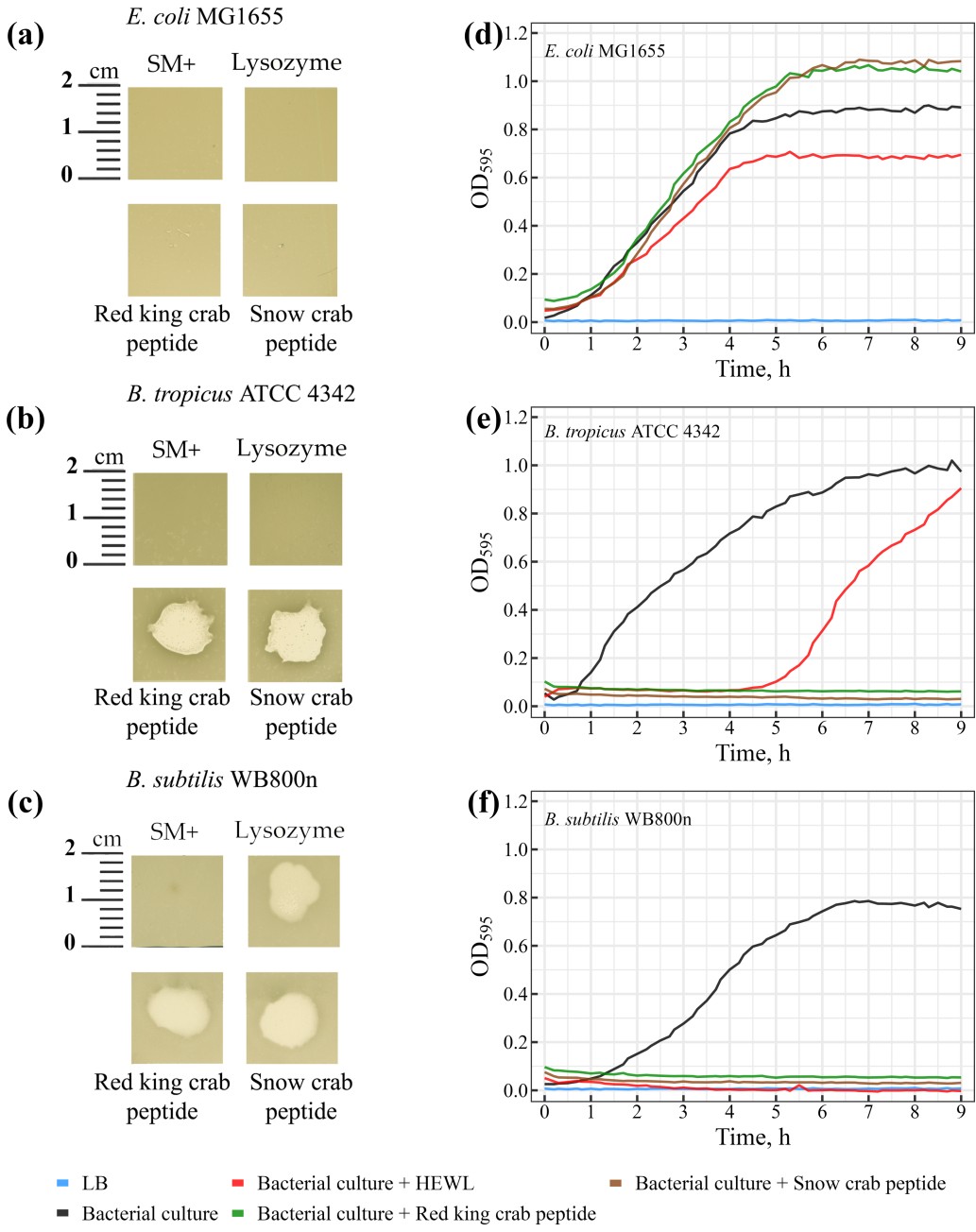

**Figure 5** (A–C) **Study of the lytic activity of peptides on bacterial strains.** The standard deviation and mean values of OD$_{595}$ are presented as CSV files in the Supplementary Materials.

It was suggested that the peptide could be retained on the Cibacron Blue 3G-A dye, which contains an anthraquinone residue that also contains three rings, two of which are aromatic. This dye is covalently linked to agarose in the chromatographic resin Blue Sepharose 6 Fast Flow. Blue Sepharose 6 Fast Flow is known to bind many proteins such as albumin, interferon, lipoproteins, coagulation factors and some enzymes including

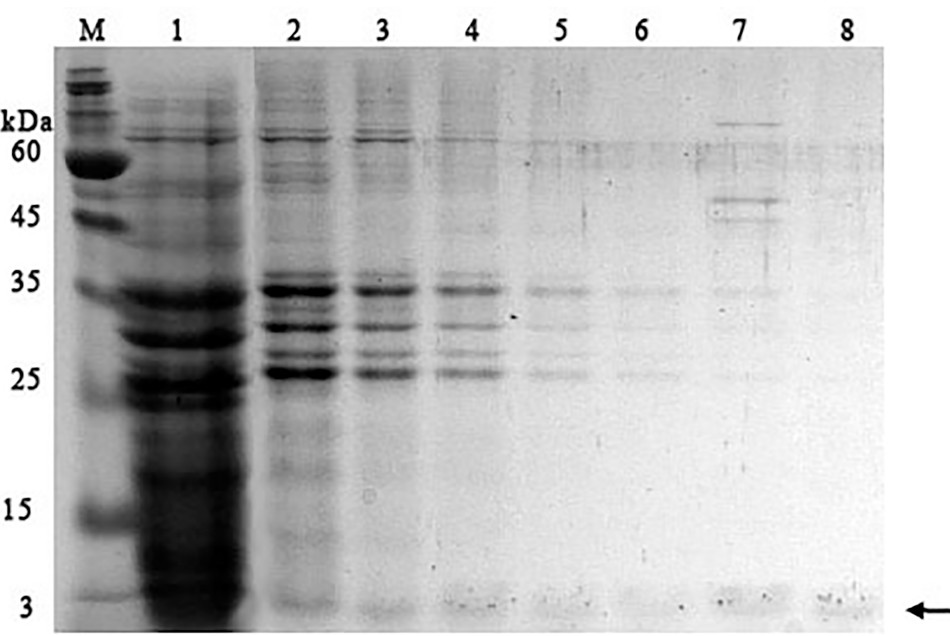

**Figure 6  Analysis of chromatographic fractions of red king crab HPC extract purified on Blue Sepharose 6 Fast Flow.** Lanes: M, molecular weight markers; 1–2, unbound proteins fractions from Blue Sepharose; 3, fraction eluted in five mM NaCl; 4, fraction eluted in 10 mM NaCl; 5, fraction eluted in 20 mM NaCl; 6, fraction eluted in 30 mM NaCl; 7, fraction eluted in 40 mM NaCl; 8, fraction eluted in 50 mM NaCl. The peptide band is shown by the arrow.

kinases, dehydrogenases and enzymes requiring adenyl-containing cofactors. We attempted to purify the peptide from the extract of low-molecular-weight protein fractions of the red king crab HPC on Blue Sepharose 6 Fast Flow. A lyophilized sample of the extract of low-molecular-weight protein fractions from the hepatopancreas was dissolved in 400 μL of loading buffer and loaded onto a column with Blue Sepharose 6 Fast Flow. Elution was performed with a NaCl gradient. The fractions were analyzed in 16% SDS-PAGE (*Laemmli, 1970*).

It was found that most of the proteins were in the fraction of unbound proteins and the column capacity was not sufficient. However, the peptide band was detected in the fractions eluted with 10–50 mM NaCl, most of which was in the 50 mM NaCl fraction (Fig. 6). Thus, the peptide was weakly retained on the Blue Sepharose 6 Fast Flow column. The weak interaction is probably due to the presence of a large amount of salt in the extract itself before loading, which was visible on the conductometric curve. Analysis of activity by zymography using a cell wall showed that the fractions with the activity were the fraction of unbound proteins and the fraction in 10 mM NaCl (Fig. 7). Since the column capacity was not sufficient in the first purification, a second purification was performed on Sepharose 6 Fast Flow for the fraction of unbound proteins (fraction 4, see Fig. 6). The fractions were analyzed in 16% SDS-PAGE. It was found that the 10 mM NaCl fraction contained almost pure peptide (Fig. 8).

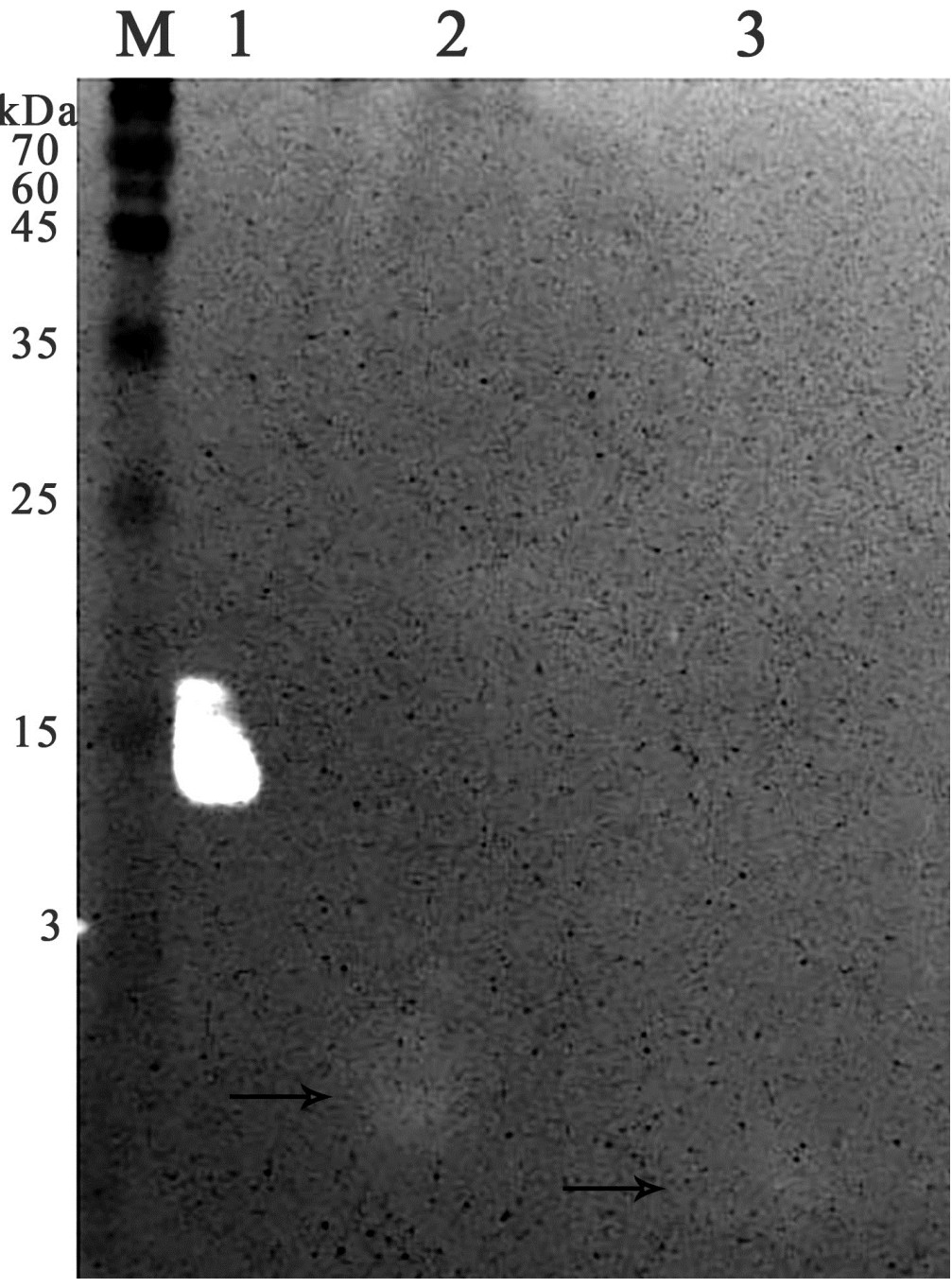

**Figure 7** **Zymographic analysis of chromatographic fractions of red king crab HPC extract purified on a Blue Sepharose 6 Fast Flow for *M. luteus* cell wall degradation.** Lanes: M, molecular weight markers; 1, HEWL; 2, unbound proteins fraction from the Blue Sepharose; 3, fraction eluted in 10 mM NaCl from the Blue Sepharose. The peptide band is shown by the arrows.

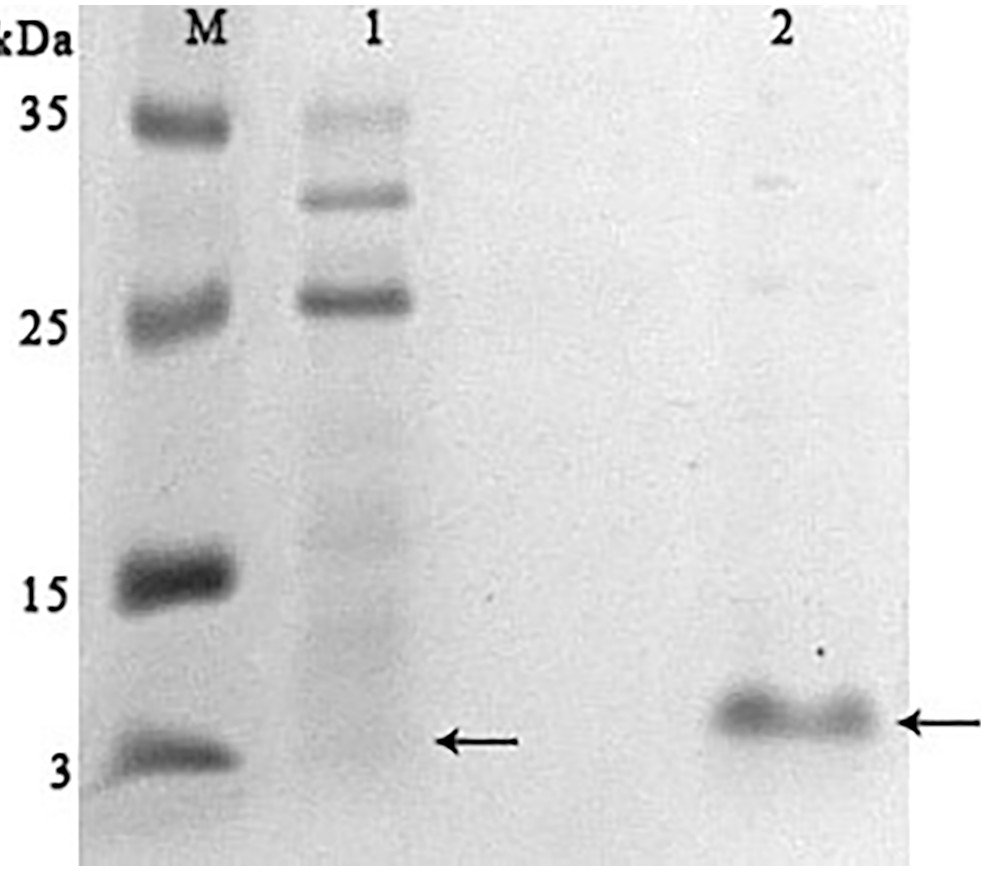

**Figure 8** **Analysis of the re-purification on the Blue Sepharose 6 Fast Flow of the unbound proteins fraction of red king crab HPC.** Lanes: M, molecular weight markers; 1, unbound protein fraction from Blue Sepharose 6 Fast Flow; 2, fraction eluted in 10 mM NaCl. The peptide band is shown by the arrows.

In parallel, to remove higher molecular weight protein contaminants from the sample of 10 mM NaCl fraction obtained during the first purification on Blue Sepharose 6 Fast Flow (fraction 4, see Fig. 6), an attempt was made to purify it on the MonoQ anion chromatography column with NaCl gradient. The fractions were analyzed in 16% SDS-PAGE (Fig. 9). It was found that this step did not lead to a significant purification from impurities, but the analysis of peptide activity towards the cell wall, carried out by the turbidimetric method (*Fukushima & Sekiguchi, 2016*), revealed that the activity of the peptide in the fraction with 10 mM NaCl increased by one and a half times compared to the fraction of unbound proteins (Fig. 10). Thus, the scheme for obtaining a pure peptide preparation from red king crab HPC can be presented as follows (Fig. 11). It should be noted that for the snow crab peptide, this scheme did not enable its complete purification. This proves that despite the common property—the cleavage of the cell wall polysaccharide (*Molchanov et al., 2023*; *Molchanov et al., 2024*)—they have different nature, which is also confirmed by the results on the effect of ionic strength and a disulfide bond reducing agent on their activity.

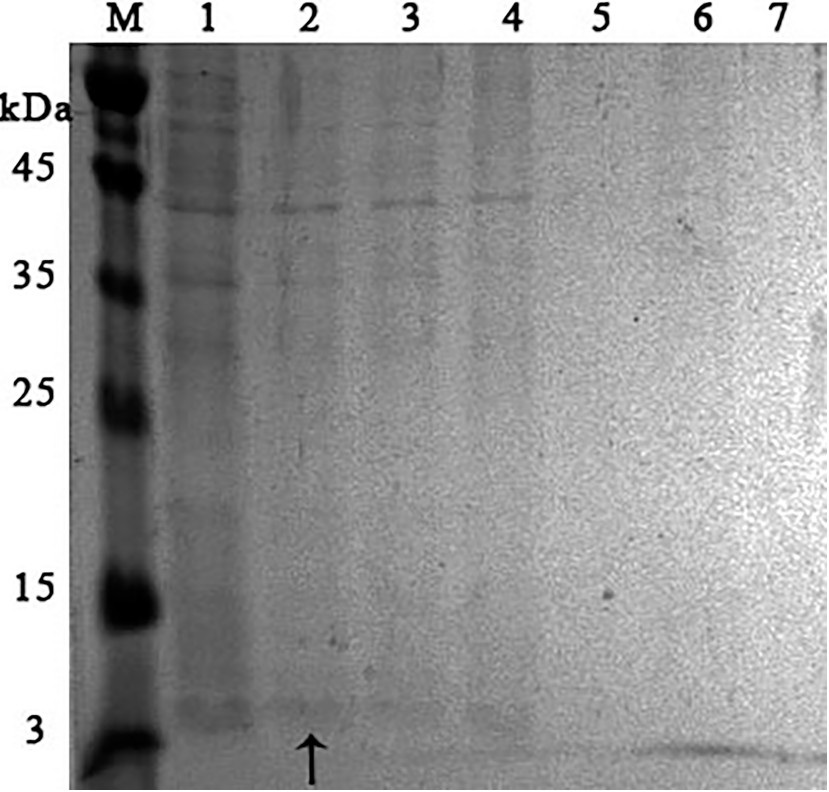

**Figure 9** **Analysis of the purification on the MonoQ column of the red king crab HPC chromatographic fraction in 10 mM NaCl obtained on Blue Sepharose 6 Fast Flow.** Lanes: M, molecular weight markers; 1, unbound proteins fraction; 2–3, fractions eluted in 10 mM NaCl; 4, fraction eluted in 30 mM NaCl; 5–7, fractions eluted in 50 mM NaCl. The peptide band is shown by the arrow.

## DISCUSSION

Antimicrobial peptides are small molecules that are components of ancient innate immunity and have unique properties that allow them to bypass bacterial cell defences. Their small size, great diversity and broad spectrum of antimicrobial activity make them an attractive basis for the creation of new antibacterial drugs, the so-called natural antibiotics. By introducing protective chemical groups into their structure or replacing individual amino acids, it is possible to improve resistance to proteases, improve target delivery and reduce cytotoxicity. Antimicrobial peptides from marine invertebrates may have exceptional properties due to their ability to survive in extreme conditions and poikilothermia, suggesting that the discovered peptides are active over a wide range of temperatures. The study of the hepatopancreas, an organ of immunity and metabolism, may be the key to understanding how the innate immune system of marine invertebrates is structured and how it differs between different infraorders, in particular between true crabs and craboids. It should be noted that the hepatopancreas is a by-product of the red king crab fishery, so its antibacterial properties may be used to develop and produce various biotechnological products based on it.

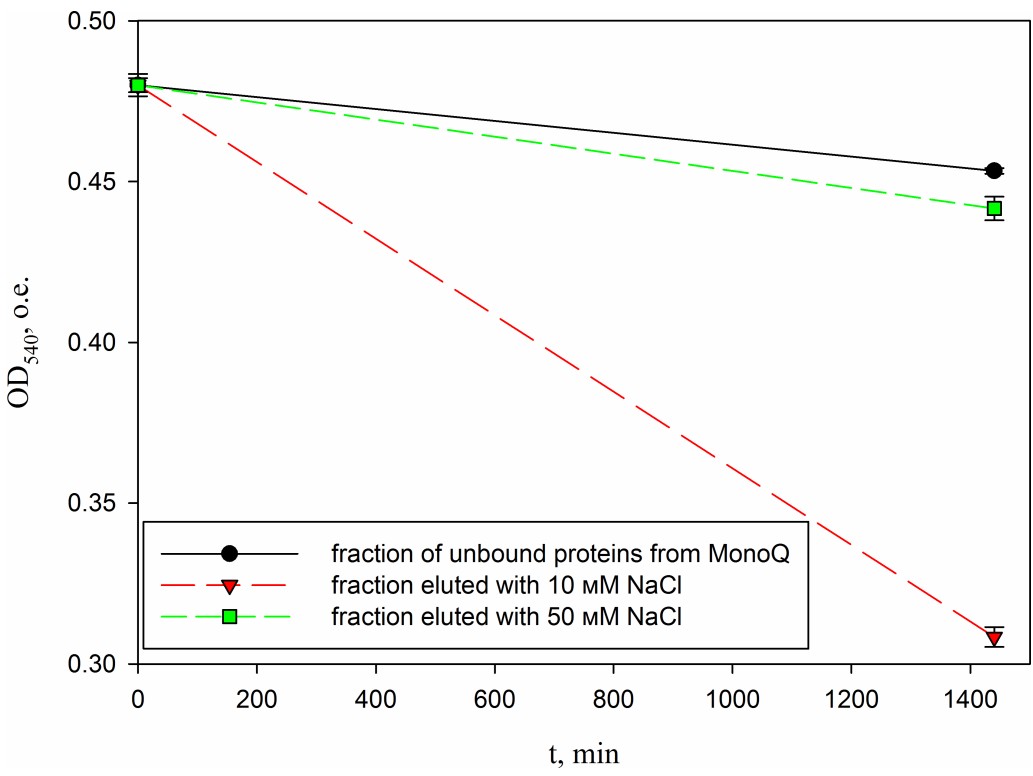

**Figure 10** **Analysis of the activity of a peptide from the red king crab HPC in the 10 mM NaCl fraction of Blue Sepharose purified on a MonoQ.** The figure shows the degradation of *M. luteus* cell wall in the presence of unbound proteins fraction and the fractions eluted with 10 and 50 mM NaCl.

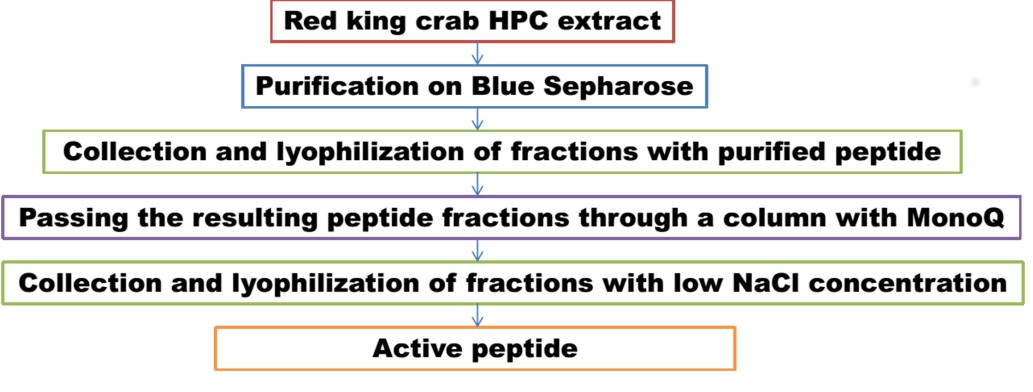

**Figure 11** Scheme of peptide purification from the red king crab HPC.

In our study, we compared the antibacterial properties of peptides found in the hepatopancreas of red king crab and snow crab. The molecular weights of the peptides differed: the red king crab peptide was about five kDa, while the snow crab peptide was about three kDa. Both peptides exhibited maximum activity at neutral and slightly alkaline pH. It was found that the peptide from the red king crab hepatopancreas was very sensitive

to salt content. This property has been found in many antimicrobial peptides. For example, the peptide antibiotics human β-defensin 1 (hBD-1) and β-defensin 2 (hBD-2), which are involved in the pathogenesis of cystic fibrosis, exhibit a same sensitivity to salt and can be inactivated in the high salt environment of the cystic fibrosis airway surface (*Bals et al., 1998*). A simple strategy to increase the salt resistance of antimicrobial peptides by replacing tryptophan or histidine residues with the bulky amino acids β-naphthylalanine and β-(4,4′-biphenyl)alanine has been described in *Yu et al., (2011)*. At the same time, a slight increase in activity in the presence of salt was observed for the snow crab peptide.

The activity of the red king crab peptide was greatly reduced in the presence of a disulfide bond reducing agent. Although the cysteine content of the red king crab peptide is low according to our preliminary mass-spectrometry data, it is likely to play an important role in its activity. The snow crab peptide was almost insensitive to the presence of DTT.

It is known that the main modes of action of antimicrobial peptides on the membrane are its destruction through the "detergent"-like mechanism, as in the carpet model, and pores, as in the barrel stave and toroidal pore models (*Huan et al., 2020*). The peptides found in the hepatopancreas of the red king crab and snow crab also acted on the cell wall by a mechanism analogous to HEWL—by degrading the polysaccharide of peptidoglycan, which we have previously shown (*Molchanov et al., 2023*; *Molchanov et al., 2024*). The peptides also exhibited similarly high antibacterial activity against gram-positive *B.tropicus* and *B.subtilis* and had no effect on gram-negative *E. coli*.

For the red king crab peptide, an approach was developed to purify it from the hepatopancreas on the Blue Sepharose 6 Fast Flow chromatographic column, probably due to its hydrophobic interactions with the Cibacron Blue 3G-A dye covalently attached to the column. For the snow crab peptide, this approach did not provide such a good purification, once again confirming the different nature of the peptides despite analogous mechanism of action and activity. Undoubtedly, further studies of the discovered peptides are required to identify their amino acid sequences, which will allow, on the basis of these data, to obtain recombinant variants for a more extensive study of the factors influencing activity and structure determination. In the future, the discovered AMPs may become the basis for the design of perspective antibacterial therapeutic agents to overcome the problem of antibiotic resistance.

## CONCLUSIONS

Thus, the study of antimicrobial peptides from red king crab and snow crab showed that they have high antibacterial activity, especially against gram-positive bacteria. Differences in molecular weight, mechanism of action and sensitivity to various factors emphasize the uniqueness of each peptide. Further studies are aimed at identifying the amino acid composition and designing recombinant forms of these peptides for wider use in medicine as antibacterial therapeutic agents.

## ACKNOWLEDGEMENTS

The authors are grateful to Azev V. for his help with cell wall and polysaccharide.

### Funding

This research was carried out within the framework of basic research on topics that correspond to State Assignment No. 075-00223-25-03 (Institute of Theoretical and Experimental Biophysics, Russian Academy of Sciences). The funders had no role in study design, data collection and analysis, decision to publish, or preparation of the manuscript.

### Grant Disclosures

The following grant information was disclosed by the authors:
Institute of Theoretical and Experimental Biophysics, Russian Academy of Sciences: 075-00223-25-03.

### Competing Interests

The authors declare there are no competing interests.

### Author Contributions

- Vladislav Molchanov conceived and designed the experiments, performed the experiments, analyzed the data, prepared figures and/or tables, and approved the final draft.
- Alfiya Yunusova performed the experiments, analyzed the data, prepared figures and/or tables, and approved the final draft.
- Olesya Kazantseva performed the experiments, analyzed the data, prepared figures and/or tables, authored or reviewed drafts of the article, and approved the final draft.
- Alexander Yegorov performed the experiments, analyzed the data, prepared figures and/or tables, and approved the final draft.
- Alexander Lukin performed the experiments, authored or reviewed drafts of the article, and approved the final draft.
- Alexander Timchenko performed the experiments, authored or reviewed drafts of the article, and approved the final draft.
- Vitaly Novikov performed the experiments, authored or reviewed drafts of the article, and approved the final draft.
- Nikolay Novojilov performed the experiments, authored or reviewed drafts of the article, and approved the final draft.
- Maria Timchenko conceived and designed the experiments, performed the experiments, analyzed the data, prepared figures and/or tables, authored or reviewed drafts of the article, and approved the final draft.

### Data Availability

The full-length uncropped gels and raw data are available in the Supplemental Files.

## Supplemental Information

Supplemental information for this article can be found online at http://dx.doi.org/10.7717/peerj.19989#supplemental-information.

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
