# Peer review of "Comparison of the antibacterial properties of peptides from the hepatopancreas of red king crab and snow crab and development of an approach for red king crab peptide isolation from the hepatopancreas"

_PeerJ, doi:10.7717/peerj.19989_

## Round 0.1 · original submission · Major Revisions

Please adress all reviewers comments.

Reviewer 1 ·

Basic reporting

The authors have a clear objective, and they have characterized and purified new AMPS that are of clinical significance.
The English language used throughout is professional, but in certain areas, it needs more clarity and professionalism, which is shared below. Research question well defined, relevant & meaningful.
INTRODUCTION & BACKGROUND:
The introduction from lines 74 to 83 has to be reworked as it does not convey exactly what the authors wish to say.
Fourth paragraph –The Mechanism and cytotoxicity to cell lines are discussed, and there is a lack of correlation.
Literature well referenced & relevant.

Experimental design

The experimental design is formulated to a high technical & ethical standard.
Methods are described with sufficient detail & information to replicate
Suggestions: Remove materials because the methods subhead is not given separately.
Add a subhead – Peptide extraction: Acetonitrile (HPLC grade) and trifluoroacetic acid (TFA) (extra pure grade) were used for peptide extraction.
Line 159: The laboratory produced samples of enzyme preparations (acetone powders), which were used for research. Is this enzyme or protein - clarify

Validity of the findings

All data have been provided; they are strong, statistically sound, & controlled.
Conclusions are well stated, linked to the original research question & limited to supporting results.
The method adopted for purification is interesting, and the results are well presented.
Figures are relevant, well labelled & described. In figures, where gel is placed, 15 and 3 kDa are not clear in purified samples when compared to the marker.
Raw data supplied
Corrections: Line -264: and Figure 2 Spelling of M. lysodeikticus should be corrected - M. lysodecticus
Line 277 change to as in Figure 2

Additional comments

The manuscript is scientifically documented and highly relevant in this era of antimicrobial resistance by microbes. The authors have done an extensive study, and the results are proven.
The raw data and supplementary files are clear and error-free. The English language can be improved so that it can enhance the quality of the article.
I commend the authors for their extensive data set and recommend the acceptance of the article after addressing the queries raised.

Reviewer 2 ·

Basic reporting

The manuscript reports the isolation of antimicrobial peptides (AMPs) from the Hepatopancreas (HPC) of red king crab and snow crab collected in Russia. This manuscript is the continuation of the previous study reported by Molchanov et al. (2023) and Molchanov et al. (2024). In this study, the AMPs were characterized based on their molecular weight, their ability to disrupt bacterial cell walls, and their activity under varying pH levels, salt concentrations, and in the presence of the disulfide-reducing agent DTT. Their antibacterial activity was also assessed using lawn assays against Escherichia coli, Bacillus cereus, and Bacillus subtilis.
The English used in the manuscript is generally sufficient; however, it could benefit from revision to improve clarity and flow. Specifically, paragraphs that consist of only one or two sentences (i.e. line 147, 152, 161, 163, 165, 178, 202, 237, 250, 289, 297, 344, 352, 374, 386, 386, 390, 420, 438, and 442) should be expanded or integrated into surrounding text. Additionally, there are some methodological descriptions currently placed in the Results section. It should be relocated to the Methods section for better organization.

Experimental design

The manuscript presents an original idea with clearly defined aims. However, the authors are advised to ensure that ethical clearance is obtained and stated when conducting research involving animal subjects.
Regarding lines 222–223 and 227–230, the manuscript refers to lysozyme and SM+ buffer as the negative and positive controls, respectively. However, in Figure 5, no activity is observed for SM+. Please be aware of the definition of positive and negative control.

Validity of the findings

The novelty of the findings aligns well with the scope and objectives of the journal. All experiments were conducted in triplicate to ensure reproducibility. However, please consider including the statistical analysis results directly in the figures to support the interpretation of the data. Do not use ambiguous words, i.e., slightly, very, or similar words.

In line 367, the authors mentioned their peptide as putative. Moreover, there are no clear indications of peptides in figures 6 and 7, any peptide characterization effort. I am not sure whether the substance in the study is new peptides without characterization and comparison with previously known peptides.
It is also recommended to include an image or an animated illustration of the hepatopancreas, along with photographs of the crabs used in the study, to enhance the readers' understanding of the experimental context.
The current figures are difficult to interpret, i.e., figures 2-5 and 6-9. The authors are encouraged to explore alternative graphical presentation methods, particularly for data with numerous points. Adding arrows or markers in Figure 6-9 to highlight the results discussed in the main text would improve clarity. The resolution and overall quality of the figures should also be enhanced to better support the study’s findings.

Reviewer 3 ·

Basic reporting

Meets standards

Experimental design

Fails on research question and methods - the structure of the peptides needs to be established.

Validity of the findings

Fails on the relevance of the anti-bacterial assays. The authors are suggesting in their conclusions that these can be used as "natural" antibiotics, but the rationale for the bacteria used needs to be clarified.

Additional comments

This manuscript by Timchenko describes the isolation and antibacterial assessment of peptides from two(?) different crab species. Both were active against Gram-positive bacteria, but the relevance of the bacteria tested is unclear - are these pathogenic bacteria that are WHO priority pathogens?
All experiments have been completed well, with appropriate discussion of the data obtained.

The most striking issue I have with the manuscript is the lack of characterisation data on the peptides discussed. Only an estimated molecular weight is given, with no evidence of amino acid composition, thus, it is hard to tell that they are peptides. While there is some indirect evidence through reduced activity from exposure of one isolated peptide to a reducing agent, this is not proof that a disulfide-linked peptide was isolated. It is also difficult to establish the purity of what is being tested, which is important when drawing conclusions about the activity.

Other comments:
1. The title is misleading; there is no isolation from the snow crab, only a comparison from the author’s earlier work (ref 25).
2. Are Figures 6,8, and 9 referring to the red king crab?
3. Line 64-saturated with disulfide bonds, is incorrect terminology. Saturated refers to the process of converting a carbon-carbon double bond into a carbon-carbon single bond, or that a solution is “saturated” so that no more analyte will dissolve. It sounds like here that the peptide has too many disulfide bonds so that no more can be added.

---

## Round 0.2 · Major Revisions

Please address the remaining concerns.

Reviewer 1 ·

Basic reporting

No comments

Experimental design

No comments

Validity of the findings

No comments

Additional comments

Since the authors have addressed the comments raised by me and is acceptable, I recommend to accept the article.

Reviewer 2 ·

Basic reporting

- I still found paragraph that contain only one sentence: line 128, 196, 307, 310, and 380. Please expand those paragraphs with some additional supporting sentences.

- Please do not change peptide with protein, or the other way around. Use the term consistently throughout the manuscript.

- Paragraph in line 243 and line 260 in the results section still have some sentences about method, they are need to be moved to methods.

Experimental design

- The authors made significant change about the figures. However, even though the subtitle in the methods section is statistical analisis, there are only SD error bar in paper. Please consider to remove the subtile of statistical analysis. Otherwise, please conduct statistical analysis on the data and then put asterix mark on the significant data.

Validity of the findings

- In this study, there is no identification that has been done for the peptide. Therefore, I suggest the authors remove the word “new antimicrobial peptides” from the paper title and other places in the manuscript and change it to “hepatopancreatic peptides”. Otherwise, the authors need to put data in the current manuscript that support their claim about novelty of their peptides.

---

## Round 0.3 · accepted · Accept

Thanks for addressing the reviewers' concerns.

Reviewer 2 ·

Basic reporting

no comment

Experimental design

no comment

Validity of the findings

no comment

Additional comments

The authors have made significant changes on the manuscript based on the reviewer's comments.